# Tensorized Embedding Layers for Efficient Model Compression

## Abstract

The embedding layers transforming input words into real vectors are the key components of deep neural networks used in natural language processing. However, when the vocabulary is large, the corresponding weight matrices can be enormous, which precludes their deployment in a limited resource setting. We introduce a novel way of parametrizing embedding layers based on the Tensor Train (TT) decomposition, which allows compressing the model significantly at the cost of a negligible drop or even a slight gain in performance. We evaluate our method on a wide range of benchmarks in natural language processing and analyze the trade-off between performance and compression ratios for a wide range of architectures, from MLPs to LSTMs and Transformers.

## 1 Introduction

Deep neural networks (DNNs) typically used in natural language processing (NLP) employ large embeddings layers, which map the input words into continuous representations and usually have the form of lookup tables. Despite such simplicity and, arguably because of it, the resulting models are cumbersome, which may cause problems in training and deploying them in a limited resource setting. Thus, the compression of large neural networks and the development of novel lightweight architectures have become essential problems in NLP research.

One way to reduce the number of parameters in the trained model is to imply a specific structure on its weight matrices (e.g., assume that they are low-rank or can be well approximated by low-rank tensor networks). Such approaches are successful at compressing the pre-trained models, but they do not facilitate the training itself. Furthermore, they usually require an additional fine-tuning stage to recover the performance of the original model.

In this paper, we introduce a new, parameter efficient embedding layer, termed TT–embedding, which can be plugged in into any model and trained end-to-end. The benefits of our compressed TT–layer are twofold. Firstly, instead of storing huge embedding matrix, we store a sequence of much smaller 2-dimensional and 3-dimensional tensors, necessary for reconstructing the required embeddings, which allows compressing the model significantly at the cost of a negligible performance drop. Secondly, the overall number of parameters can be relatively small (and constant) during the whole training stage, which allows to use larger batches or train efficiently in a case of limited resources.

To validate the efficiency of the proposed approach, we have tested it on several popular NLP tasks. In our experiments, we have observed that the standard embeddings can be replaced by TT–embeddings with the compression ratio of $1 - 3$ orders without any significant drop (and sometimes even with a slight gain) of the metric of interest. Specifically, we report the following compression ratios of the embedding layers: $441$ on the IMDB dataset with $0.2\%$ absolute increase in classification accuracy; $15$ on the WMT 2014 En–De dataset with $0.3$ drop in the BLEU score.

Additionally, we have also evaluated our algorithm on a task of binary classification based on a large number of categorical features. More concretely, we applied TT–embedding to the click through rate (CTR) prediction problem, a crucial task in the field of digital advertising. Neural networks, typically used for solving this problem, while being rather elementary, include a large number of embedding layers of significant size. As a result, a majority of model parameters that represent these layers, may occupy hundreds of gigabytes of space. We show that TT–embedding not only considerably reduces the number of parameters in such models, but also sometimes improves their accuracy.

## 2 RELATED WORK

In recent years, a large body of research was devoted to compressing and speeding up various components of neural networks used in NLP tasks. Joulin et al. (2016) adapted the framework of product quantization to reduce the number of parameters in linear models used for text classification. See et al. (2016) proposed to compress LSTM-based neural machine translation models with pruning algorithms. Lobacheva et al. (2017) showed that the recurrent models could be significantly sparsified with the help of variational dropout (Kingma et al., 2015). Chen et al. (2018b) proposed more compact K-way D-dimensional discrete encoding scheme to replace the "one-hot" encoding of categorical features, such as words in NLP taks. Very recently, Chen et al. (2018a) and Variani et al. (2018) introduced GroupReduce and WEST, two very efficient compression methods for the embedding and softmax layers, based on structured low-rank matrix approximation. Concurrently, Lam (2018) proposed the quantization algorithm for compressing word vectors and showed the superiority of the obtained embeddings on word similarity, word analogy, and question answering tasks.

Tensor methods have also been already successfully applied to neural networks compression. Novikov et al. (2015) coined the idea of reshaping weights of fully-connected layers into high-dimensional tensors and representing them in Tensor Train (TT) (Oseledets, 2011) format. This approach was later extended to convolutional (Garipov et al., 2016) and recurrent (Yang et al., 2017; Tjandra et al., 2017; Yu et al., 2017) neural networks. Furthermore, Lebedev et al. (2015) showed that convolutional layers could be also compressed with canonical (CP) tensor decomposition (Carroll & Chang, 1970; Harshman, 1970). Finally, Wang et al. (2018) compressed both fully-connected and convolutional layers with Tensor Ring decomposition (Zhao et al., 2016). While all these methods allowed to reduce the number of parameters in the networks dramatically, they mostly capitalized on heavy fully-connected and convolutional layers (present in AlexNet (Krizhevsky et al., 2012) or VGG (Simonyan & Zisserman, 2014)), which became outdated in the following years. Recently, Ma et al. (2019) succesfully applied Block-Term Tensor Decomposition to the compression of self-attention modules in the Transformer (Vaswani et al., 2017) architecture. In this work, we show the benefits of applying tensor machinery to the compression of embedding layers, which are still widely used in NLP.

## 3 TENSOR TRAIN EMBEDDING

In this section, we briefly introduce the necessary notation and present the algorithm for training the TT–embedding layer. Hereinafter, by $N$-way tensor $\boldsymbol{\mathcal{X}}$ we mean a multidimensional array:

$$\boldsymbol{\mathcal{X}} \in \mathbb{R}^{I_1 \times I_2 \times \cdots \times I_N}.$$

with entries $\boldsymbol{\mathcal{X}}(i_1, \ldots, i_N)$, such that $\{0 \le i_k < I_k\}_{k=1}^N$.

### 3.1 MOTIVATION

Since most of the parameters in the NLP models occupy the embedding layers, we can greatly reduce size of the entire model by compressing these layers. Our goal is to replace the standard embedding matrix with a more compact, yet powerful and trainable, representation which would allow us to efficiently map words into vectors.

The simplest approach to compactly represent a matrix of a large size is to use the low–rank matrix factorization, which treats matrix $\mathbf{E} \in \mathbb{R}^{I \times J}$ as a product of two matrices $\mathbf{E} = \mathbf{U}\mathbf{V}^\top$. Here $\mathbf{U} \in \mathbb{R}^{I \times R}$ and $V \in \mathbb{R}^{J \times R}$ are much "thinner" matrices, and $R$ is the rank hyperparameter. Note that rather than training the model with the standard embedding layer, and then trying to compress the obtained embedding, we can initially seek the embedding matrix in the described low–rank format. Then, for evaluation and training, the individual word embedding $\mathbf{E}[i, :]$ can be computed as a product $\mathbf{U}[i, :]\mathbf{V}^\top$ which does not require materializing the full matrix $\mathbf{E}$. This approach reduces the number of degrees of freedom in the embedding layer from $IJ$ to $(I + J)R$.

However, typically, in the NLP tasks the embedding dimension $J$ is much smaller than the vocabulary size $I$, and obtaining significant compression ratio using low-rank matrix factorization is problematic. In order to preserve the model performance, the rank $R$ cannot be taken very small, and the compression ratio is bounded by $\frac{IJ}{(I+J)R} \le \frac{J}{R}$, which is close to $1$ for usually full-rank embedding matrix (see Figure 1 in Chen et al. (2018b)). To overcome this bound and achieve significant compression

ratio even for matrices of disproportional dimensionalities, we reshape them into multidimensional tensors and apply the *Tensor Train* decomposition, which allows for more compact representation, where the number of parameters falls down to logarithmic with respect to $I$.

## 3.2 TENSOR TRAIN DECOMPOSITION

A tensor $\boldsymbol{\mathcal{X}}$ is said to be represented in the Tensor Train (TT) format (Oseledets, 2011) if each element of $\boldsymbol{\mathcal{X}}$ can be computed as:

$$\boldsymbol{\mathcal{X}}(i_1, i_2, \ldots, i_d) = \sum_{r_1=1}^{R_1} \sum_{r_2=1}^{R_2} \cdots \sum_{r_{N-1}=1}^{R_{N-1}} \boldsymbol{\mathcal{G}}^{(1)}(i_1, r_1) \boldsymbol{\mathcal{G}}^{(2)}(r_1, i_2, r_2) \ldots \boldsymbol{\mathcal{G}}^{(N)}(r_{N-1}, i_N),$$

where the tensors $\boldsymbol{\mathcal{G}}^{(k)} \in \mathbb{R}^{R_{k-1} \times I_k \times R_k}$ are the so-called *TT–cores* and $R_0 = R_N = 1$ by definition. The minimal values of $\{R_k\}_{k=1}^{N-1}$ for which the TT–decomposition exists are called *TT–ranks*. Note, that the element $\boldsymbol{\mathcal{X}}(i_1, i_2 \ldots i_N)$ is just effectively the product of 2 vectors and $N - 2$ matrices:

$$\boldsymbol{\mathcal{X}}(i_1, \ldots, i_N) = \underbrace{\boldsymbol{\mathcal{G}}^{(1)}[i_1, :]}_{1 \times R_1} \underbrace{\boldsymbol{\mathcal{G}}^{(2)}[:, i_2, :]}_{R_1 \times R_2} \ldots \underbrace{\boldsymbol{\mathcal{G}}^{(N-1)}[:, i_{N-1}, :]}_{R_{N-2} \times R_{N-1}} \underbrace{\boldsymbol{\mathcal{G}}^{(N)}[:, i_N]}_{R_{N-1} \times 1},$$

where $\boldsymbol{\mathcal{G}}^{(k)}[:, i_k, :]$ stands for the slice (a subset of a tensor with some indices fixed) of the corresponding TT–core $\boldsymbol{\mathcal{G}}^{(k)}$.

The number of degrees of freedom in such a decomposition can be evaluated as $\sum_{k=1}^{N} R_{k-1} I_k R_k$. Thus, in the case of small ranks, the total number of parameters required to store a tensor in TT–representation is significantly smaller than $\prod_{k=1}^{N} I_k$ parameters required to store the full tensor of the corresponding size. This observation makes the application of the TT–decomposition appealing in many problems dealing with extremely large tensors.

## 3.3 TT–MATRIX

Let $\mathbf{X} \in \mathbb{R}^{I \times J}$ be a matrix of size $I \times J$. Given two arbitrary factorizations of its dimensions into natural numbers, $I = \prod_{k=1}^{N} I_k$ and $J = \prod_{k=1}^{N} J_k$, we can reshape[1] and transpose this matrix into an $N$-way tensor $\boldsymbol{\mathcal{X}} \in \mathbb{R}^{I_1 J_1 \times I_2 J_2 \times \cdots \times I_N J_N}$ and then apply the TT–decomposition to it, resulting in a more compact representation.

More concretely, define the bijections $\boldsymbol{\mathcal{I}}(i) = (i_1, \ldots, i_N)$ and $\boldsymbol{\mathcal{J}}(j) = (j_1, \ldots, j_N)$ that map row and column indices $i$ and $j$ of the matrix $X$ to the $N$-dimensional vector-indices such that $0 \le i_k < I_k$, $0 \le j_k < J_k$, $\forall k = 1, \ldots, N$. From the matrix $\mathbf{X}$ we can form an $N$-way tensor $\boldsymbol{\mathcal{X}}$ whose $k$-th dimension is of length $I_k J_k$ and is indexed by the tuple $(i_k, j_k)$. This tensor is then represented in the TT–format:

$$\boldsymbol{\mathcal{X}}((i_1, j_1) \ldots (i_N, j_N)) = \boldsymbol{\mathcal{G}}^{(1)}[(i_1, j_1), :] \ldots \boldsymbol{\mathcal{G}}^{(N)}[:, (i_N, j_N)]. \tag{1}$$

Such representation of the matrix in the TT–format is called *TT–matrix* (Oseledets, 2010; Novikov et al., 2015) and is also known as Matrix Product Operator (Pirvu et al., 2010) in physics literature. The factorizations $(I_1, I_2, \ldots I_N) \times (J_1, J_2, \ldots J_N)$ will be referred to as the *shape* of TT–matrix, or *TT–shapes*. The process of constructing the TT–matrix from the standard matrix is visualized in Figure 1 for the tensor of order 3. Note, that in this case the TT–cores are in fact 4-th order tensors, but all the operations defined for tensors in the TT–format are naturally extended to TT–matrices.

## 3.4 TT–EMBEDDING

By *TT–embedding*, we call a layer with trainable parameters (TT–cores) represented as a TT–matrix $\boldsymbol{\mathcal{E}}$ of the underlying tensor shape $(I_1, I_2, \ldots I_N) \times (J_1, J_2, \ldots J_N)$, which can be transformed into a valid embedding layer $E \in \mathbb{R}^{I \times J}$, with $I = \prod_{k=1}^{N} I_k$ and $J = \prod_{k=1}^{N} J_k$. To specify the shapes of TT–cores one has also to provide the TT–ranks, which are treated as hyperparameters of the layer and explicitly define the total compression ratio.

---

[1]by reshape we mean a column-major `reshape` command such as `numpy.reshape` in `Python`.

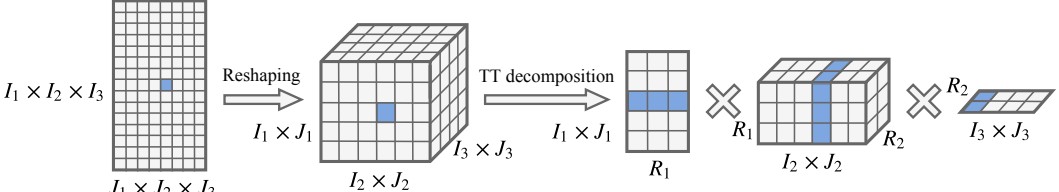

Figure 1: Construction of the TT–matrix from the standard embedding matrix. Blue color depicts how the single element in the initial matrix is transformed into the product of the highlighted vectors and matrices in the TT–cores.

In order to compute the embedding for a particular word indexed $i$ in the vocabulary, we first map the row index $i$ into the $N$-dimensional vector index $(i_1, \ldots, i_N)$, and then calculate components of the embedding with formula (1). Note, that the computation of all its components is equivalent to selecting the particular slices in TT-cores (slices of shapes $J_1 \times R_1$ in $\boldsymbol{\mathcal{G}}^{(1)}$, $R_1 \times J_2 \times R_2$ in $\boldsymbol{\mathcal{G}}^{(2)}$ and so on) and performing a sequence of matrix multiplications, which is executed efficiently in modern linear algebra packages, such as BLAS. Pseudocode for the procedure of computing the mapping $i \to (i_1, \ldots, i_N)$ is given in Appendix A.

In order to construct TT–embedding layer for a vocabulary of size $I$ and embedding dimension $J$, and to train a model with such a layer, one has to perform the following steps.

- Provide factorizations of $I$ and $J$ into factors $I = I_1 \times I_2 \times \cdots \times I_N$ and $J = J_1 \times J_2 \times \cdots \times J_N$, and specify the set of TT–ranks $\{R_1, R_2, \ldots, R_{N-1}\}$.

- Initialize the set of parameters of the embedding $\boldsymbol{\Theta} = \{\boldsymbol{\mathcal{G}}^{(k)} \in \mathbb{R}^{R_{k-1} \times I_k \times J_k \times R_k}\}_{k=1}^N$. Concrete initialization scenarios are discussed further in the text.

- During training, given a batch of indices $\{i_1, i_2, \ldots i_b\}$, compute the corresponding embeddings $\{\mathbf{e}_1, \mathbf{e}_2, \ldots, \mathbf{e}_b\}$ using Eq. (1) and Algorithm 1.

- Computed embeddings can be followed by any standard layer such as LSTM (Hochreiter & Schmidhuber, 1997) or self-attention (Vaswani et al., 2017), and trained with backpropagation since they differentially depend on the parameters $\boldsymbol{\Theta}$.

TT–embedding implies a specific structure on the order of tokens in the vocabulary (the order of rows in the embedding matrix), and determining the optimal order is an appealing problem to solve. However, we leave this problem for future work and use the order produced by the standard tokenizer (sorted by frequency) in our current experiments.

We also experimented with a more general form of TT-decomposition, namely Tensor Ring decomposition (Zhao et al., 2016; Wang et al., 2018). This decomposition by construction has the appealing property of being circular permutation invariant (and, thus, more robust with respect to the order of the tokens), which could have potentially provided an improvement over the TT-based models with simple frequency based ordering. Our experiments with TR decomposition on Transformer for NMT can be found in Appendix B.

**Initialization** The standard way to initialize an embedding matrix $\mathbf{E} \in \mathbb{R}^{I \times J}$ is via, e.g., Glorot initializer (Glorot & Bengio, 2010), which initializes each element as $\mathbf{E}(i, j) \sim \mathcal{N}\left(0, \frac{2}{I+J}\right)$. For the TT–embedding, we can only initialize the TT–cores, and the distribution of the elements of the resulting matrix $\boldsymbol{\mathcal{E}}$ is rather non–trivial. However, it is easy to verify that if we initialize each TT–core element as $\boldsymbol{\mathcal{G}}^{(k)}(r_{k-1}, i_k, r_k) \sim \mathcal{N}(0, 1)$, the resulting distribution of the matrix elements $\boldsymbol{\mathcal{E}}(i, j)$ has the property that $\mathbb{E}[\boldsymbol{\mathcal{E}}(i, j)] = 0$ and $\mathrm{Var}[\boldsymbol{\mathcal{E}}(i, j)] = \prod_{k=1}^N R_k = R^2$. Capitalizing on this observation, in order to obtain the desired variance $\mathrm{Var}[\boldsymbol{\mathcal{E}}(i, j)] = \sigma^2$ while keeping $\mathbb{E}[\boldsymbol{\mathcal{E}}(i, j)] = 0$, we can simply initialize each TT–core as

$$\boldsymbol{\mathcal{G}}^{(k)}(r_{k-1}, i_k, r_k) \sim \mathcal{N}\left(0, \left(\frac{\sigma}{R}\right)^{2/N}\right). \tag{2}$$

The resulting distribution is not Gaussian, however, it approaches the Gaussian distribution with the increase of the TT–rank (Figure 2).

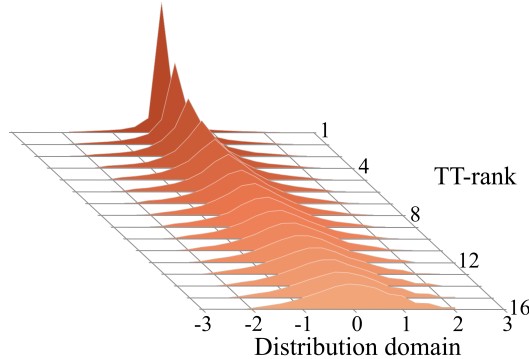

Figure 2: Distribution of matrix elements of the TT–matrix of shape $(5, 5, 5, 5) \times (5, 5, 5, 5)$ initialized by formula (2) with $\sigma = 1$. As the TT–rank increases, the resulting distribution approaches $\mathcal{N}(0, 1)$.

In our experiments, we have used the modified Glorot initializer implemented by formula (2), which greatly improved performance, as opposed to initializing TT–cores simply via a standard normal distribution. It is also possible to initialize TT–embedding layer by converting the learned embedding matrix into TT–format using the TT–SVD algorithm (Oseledets, 2011), however, this approach requires the pretrained embedding matrix and does not exhibit better performance in practice.

**Hyperparameter selection**   Our embedding layer introduces two additional structure-specific hyperparameters, namely *TT–shapes* and *TT–ranks*.

TT–embedding does not require the vocabulary size $I$ to be represented *exactly* as the product of factors $I_1, \ldots, I_N$, in fact, any factorization $\prod_{k=1}^{k} I_k = \widetilde{I} \geq I$ will suffice. However, in order to achieve the highest possible compression ratio for a fixed value of $\widetilde{I}$, the factors $\{I_k\}_{k=1}^{N}$ should be as close to each other as possible. Our implementation includes a simple automated procedure for selecting a good values of $\{I_k\}_{k=1}^{N}$ during TT–embedding initialization. The factors $J_1, \ldots, J_N$ are defined by the embedding dimensionality $J$ which can be easily chosen to support good factorization, e.g., $512 = 8 \times 8 \times 8$ or $480 = 6 \times 5 \times 4 \times 4$.

The values of TT–ranks directly define the compression ratio, so choosing them to be too small or too large will result into either significant performance drop or little reduction of the number of parameters. In our experiments, we set all TT–ranks to 16 for the problems with small vocabularies and $64 - 192$ for the problems with larger vocabularies, which allowed us to achieve significant compression of the embedding layer, at the cost of a tiny sacrifice in the metrics of interest.

## 4   EXPERIMENTS

**Code**   We have implemented TT–embeddings described in Section 3 in `Python` using `PyTorch` (Paszke et al., 2017). The code is available at the anonymous repository https://github.com/tt-embedding/tt-embeddings.

**Experimental setup**   We tested our approach on several popular NLP tasks:

- **Sentiment analysis** — as a starting point in our experiments, we test TT–embeddings on a rather simple task of predicting polarity of a sentence.

- **Neural Machine Translation (NMT)** — to verify the applicability of TT–embeddings in more practical problems, we test it on a more challenging task of machine translation.

- **Language Modeling (LM)** — then, we evaluate TT–embeddings on language modeling tasks in the case of extremely large vocabularies.

- **Click Through Rate (CTR) prediction** — finally, we show that TT–embeddings can be applied for the binary classification with categorical features of significant cardinality.

To prove the generality and wide applicability of the proposed approach, we tested it on various architectures, such as MLPs (CTR), LSTMs (sentiment analysis), and Transformers (NMT, LM).

Note that Transformers in LM and NMT use the same weight matrix for their embedding and softmax layers (Press & Wolf, 2016; Inan et al., 2016) which already significantly reduces model size. Untying weights and tensorizing the embedding layer only will lead to the increase in the number of parameters instead of compression. In our experiments, we use two separate TT-decompositions of the same shape for embedding and softmax layers and report the compression ratios as $\frac{|V| \times d_{\text{model}}}{2 \times \text{TT-params}}$.

## 4.1 SENTIMENT ANALYSIS

For this experiment, we have used the IMDB dataset (Maas et al., 2011) with two categories, and the Stanford Sentiment Treebank (SST) (Socher et al., 2013) with five categories. We have taken the most frequent 25000 words for the IMDB dataset and 17200 for SST, embedded them into a $J$–dimensional space using either standard embedding or TT–embedding layer, and performed classification using a standard bidirectional two–layer LSTM with hidden size $h = 128$, and dropout rate $P_{\text{drop}} = 0.5$.

Our findings are summarized in Table 1. We observe that the models with largely compressed embedding layers can perform equally or even better than the full uncompressed models. This suggests that learning individual independent embeddings for each particular word is superfluous, as the expressive power of LSTM is sufficient to make use of these intertwined, yet more compact embeddings. Moreover, slightly better test accuracy of the compressed models in certain cases (e.g., for the SST dataset of a rather small size) insinuates that imposing specific tensorial low–rank structure on the embedding matrix can be viewed as a special form of *regularization*, thus potentially improving model generalization. A detailed and comprehensive test of this hypothesis goes beyond the scope of this paper, and we leave it for future work.

Table 1: Sentiment analysis, LSTM on IMDB and SST datasets. Embedding compression is calculated as the ratio between the number of parameters in the full embedding layer and TT–embedding layer. The LSTM parts are identical in both models, and the TT–ranks were set to 16 in these experiments.

| Dataset | Model | Embedding shape | Test acc. | Emb compr. | Total params |
|---------|-------|-----------------|-----------|------------|--------------|
| IMDB | Full | $25000 \times 256$ | 0.886 | 1 | 7.19M |
| | TT1 | $(25, 30, 40) \times (4, 8, 8)$ | 0.871 | 93 | 0.86M |
| | TT2 | $(10, 10, 15, 20) \times (4, 4, 4, 4)$ | 0.888 | 232 | 0.82M |
| | TT3 | $(5, 5, 5, 5, 6, 8) \times (2, 2, 2, 2, 4, 4)$ | **0.897** | 441 | 0.81M |
| SST | Full | $17200 \times 256$ | 0.374 | 1 | 5.19M |
| | TT1 | $(24, 25, 30) \times (4, 8, 8)$ | **0.415** | 78 | 0.85M |
| | TT2 | $(10, 10, 12, 15) \times (4, 4, 4, 4)$ | 0.411 | 182 | 0.82M |
| | TT3 | $(4, 5, 5, 5, 6, 6) \times (2, 2, 2, 2, 4, 4)$ | 0.399 | 307 | 0.81M |

## 4.2 NEURAL MACHINE TRANSLATION

For this experiment, we have trained the Transformer-big model ($d_{\text{model}} = 1024$, $d_{\text{ff}} = 4096$, $h = 16$) from (Vaswani et al., 2017) on WMT 2014 English–German dataset consisting of roughly 4.5 million sentence pairs. We evaluated on newstest2014 dataset using beam search with a beam size of 4 and no length penalty. We did not employ checkpoint averaging and used the last checkpoint to compute the BLEU score. Sentences were tokenized with YouTokenToMe[2] byte-pair-encodings, resulting in a joint vocabulary of 32768 tokens. For the full list of hyperparameters, see Appendix C.

Our results are summarized in Table 2. We observe that even in this rather challenging task, both embedding and softmax layers can be compressed significantly, at the cost of a small drop in the

---

[2]https://github.com/VKCOM/YouTokenToMe

BLEU score. However, with the increase of compression factor, the performance deteriorates rapidly. Compared to the sentiment analysis, NMT is a much more complex task which benefits more from additional capacity (in the form of more powerful RNN or more transformer blocks) rather than regularization (Bahdanau et al., 2014; Vaswani et al., 2017; Wu et al., 2019), which may explain why we did not manage to improve the model by regularizing its embedding layers.

TT-embeddings induce $8\%$ training iteration time overhead if compared to the baseline Transformer-big due to our current implementation heavy relying on slow `torch.einsum` function while standard embedding and softmax layers make use of fast and highly-optimized Tensor Cores for mixed-precision training. We expect a dedicated CUDA kernel to be much more efficient.

Table 2: NMT, Transformer-big on WMT'14 English-to-German dataset. Both case-sensitive tokenized BLEU and de-tokenized SacreBLEU (Post, 2018) on newstest2014 are reported.

| Model | Embedding shape | TT rank | Token BLEU | Sacre BLEU | Emb compr. | Total params | Iter time |
|---|---|---|---|---|---|---|---|
| Big | $32768 \times 1024$ | — | **29.58** | **28.84** | 1 | 210M | 1.14 |
| Big+TT1 | $(32, 32, 32) \times (8, 8, 16)$ | 64 | 29.17 | 28.53 | 15.3 | 179M | 1.23 |
| Big+TT2 | $(32, 32, 32) \times (8, 8, 16)$ | 48 | 28.53 | 27.97 | 26.8 | 178M | 1.22 |
| Big+TT3 | $(32, 32, 32) \times (8, 8, 16)$ | 32 | 28.26 | 27.70 | 58.5 | 177M | 1.22 |

### 4.3 LANGUAGE MODELING

We took the Transformer-XL (Dai et al., 2019), an open source[3] state-of-the-art language modeling architecture at the time of this writing, and replaced its embedding and softmax layers with TT–factorizations. Then, we tested different model configurations on the WikiText–103 (Merity et al., 2016) dataset and reported the results in Table 3. For the full list of hyperparameters, see Appendix C.

Compared to sentiment analysis and NMT, we were not able to achieve that high compression ratios for embedding and softmax layers in LM. However, even moderate 3.8 times compression allowed us to save 100M of weights at the cost of $\sim 1.5$ perplexity drop.

Table 3: LM, Transformer-XL (Dai et al., 2019) on WikiText-103 dataset.

| Model | Embedding shape | TT rank | Valid PPL | Test PPL | Emb compr. | Total params |
|---|---|---|---|---|---|---|
| TXL | $267735 \times 512$ | — | **22.55** | **24.37** | 1 | 192M |
| TXL+TT1 | $(60, 60, 75) \times (8, 8, 8)$ | 192 | 24.38 | 25.67 | 3.8 | 94M |
| TXL+TT2 | $(60, 60, 75) \times (8, 8, 8)$ | 128 | 25.53 | 26.73 | 8.6 | 73M |
| TXL+TT3 | $(60, 60, 75) \times (8, 8, 8)$ | 96 | 26.73 | 28.04 | 15.1 | 65M |

### 4.4 CLICK THROUGH RATE PREDICTION

Among other applications of the TT–embedding layer, we chose to focus on CTR prediction, a popular task in digital advertising (He et al., 2014). We consider open dataset provided by Criteo for Kaggle Display Advertising Challenge (Criteo Labs, 2014) which consists of 39 categorical features, 45.8M samples and is binary labeled according to whether the user clicked on the given advertisement. Unique values of categorical features are bijectively mapped into integers. To reduce the memory footprint, if the size of a corresponding vocabulary is immense (e.g., a cardinality of some features in this dataset is of order $10^6$), these integers are further hashed by taking modulus with respect to some fixed number such as $10^5$. However, due to strong compression properties of TT–embeddings, this is not necessary for our approach, and we consider both full and hashed datasets in our experiments.

---

[3]https://github.com/kimiyoung/transformer-xl

**CTR with the baseline algorithm**  The task at hand can be treated as a binary classification problem. As a baseline algorithm, we consider the neural network with the following architecture. First, each of the categorical features is passed through a separate embedding layer with embedding size $J$. After that, the embedded features are concatenated and passed through 4 fully-connected layers of 1024 neurons and ReLU activation functions. In all experiments, we used Adam optimizer with the learning rate equal to 0.0005. Since many input features have a large number of unique values (e.g., 10131227) and storing the corresponding embedding matrices would be costly, we employ the hashing procedure mentioned earlier.

**CTR with TT–embeddings**  We substitute the embedding layers with the TT–embedding layers. Besides that, we leave the overall structure of the neural network unchanged with the same parameters as in the baseline approach. Table 4 presents the experimental results on the Criteo CTR dataset. To the best of our knowledge, our loss value is very close to the state-of-the-art result (Juan et al., 2016). These experiments indicate that the substitution of large embedding layers with TT–embeddings leads to significant compression ratios (up to 2011 times) with a slight improvement in the test loss, and up to 4200 with a small drop in the test loss. The total size of the compressed model does not exceed 20 Mb, while the baseline model weighs about 160 Mb. The obtained compression ratio suggests that the usage of TT–embedding layers may be beneficial in CTR prediction tasks.

Table 4: CTR prediction. The hashed dataset is constructed as specified in Section 4.4 with hashing value $10^5$. Embedding layers with more than 2000 unique tokens were replaced by TT–embeddings with shape factorizations consisting of 3 or 4 factors.

| Hash | Model | Factorization | TT rank | Hidden size | Test loss | Emb. compr. | Total params |
|------|-------|---------------|---------|-------------|-----------|-------------|--------------|
| | Full | — | — | 1024 | 0.4440 | 1 | 41.2M |
| | TT1 | 3 factors | 16 | 1024 | **0.4433** | 61 | 4.7M |
| $10^5$ | TT2 | 4 factors | 16 | 1024 | 0.4440 | 92 | 4.5M |
| | TT3 | 3 factors | 2 | 128 | 0.4515 | 2100 | 0.53M |
| | TT4 | 4 factors | 2 | 128 | 0.4530 | 4193 | 0.53M |
| — | TT1 | 3 factors | 16 | 1024 | 0.4444 | 1004 | 5.2M |
| | TT2 | 4 factors | 16 | 1024 | **0.4438** | 2011 | 4.7M |

## 5 DISCUSSION AND FUTURE WORK

We propose a novel embedding layer, the TT–embedding, for compressing huge lookup tables used for encoding categorical features of significant cardinality, such as the index of a token in natural language processing tasks. The proposed approach, based on the TT–decomposition, experimentally proved to be effective, as it heavily decreases the number of training parameters at the cost of a small deterioration in performance. In addition, our method can be easily integrated into any deep learning framework and trained via backpropagation, while capitalizing on reduced memory requirements and increased training batch size.

Our experimental results suggest several appealing directions for future work. First of all, TT–embeddings impose a concrete tensorial low-rank structure on the embedding matrix, which was shown to improve the generalization ability of the networks acting as a regularizer. The properties and conditions of applicability of this regularizer are subject to more rigorous analysis. Secondly, unlike standard embedding, we can introduce non-linearity into TT-cores to improve their expressive power (Khrulkov et al., 2019). Additionally, it is important to understand how the order of tokens in the vocabulary affects the properties of the networks with TT–embedding. We hypothesize that there exists the optimal order of tokens which better exploits the particular structure of TT–embedding and leads to a boost in performance and/or compression ratio. Finally, the idea of applying higher–order tensor decompositions to reduce the number of parameters in neural nets is complementary to more traditional methods such as pruning (Han et al., 2015) and quantization (Hubara et al., 2017; Xu et al., 2018). Thus, it would be interesting to make a thorough comparison of all these methods and investigate whether their combination may lead to even stronger compression.

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

## A    MULTIINDEX CONSTRUCTION

**Algorithm 1** The algorithm implementing the bijection $\mathcal{I}(i)$ as described in Section 3.3.

**Require**: $I$ – vocabulary size, $\{I_k\}_{k=1}^N$ – an arbitrary factorization of $I$, $i$ – index of the target word in vocabulary.
**Returns**: $\mathcal{I}(i) = (i_1, \ldots, i_N)$ – $N$-dimensional index.
**Initialize**: $L = \{1, I_1, I_1 I_2, \ldots, I_1 I_2 \ldots I_{N-1}\}$
**for** $k = N$ **to** $1$ **do**
  $i_k \leftarrow \texttt{floor}(i/L[k])$
  $i \leftarrow i \mod L[k]$
**end for**

**Algorithm 2** The algorithm implementing the bijection $(i_1, \ldots, i_N) \rightarrow i$, inverse to $\mathcal{I}(i)$.

**Require**: $I$ – vocabulary size, $\{I_k\}_{k=1}^N$ – an arbitrary factorization of $I$, $(i_1, \ldots, i_N)$ – $N$-dimensional index.
**Returns**: $i$ – index of the target word in vocabulary
**Initialize**: $L = \{1, I_1, I_1 I_2, \ldots, I_1 I_2 \ldots I_{N-1}\}$
$i \leftarrow 0$
**for** $k = 1$ **to** $N$ **do**
  $i \leftarrow i + i_k \times L[k]$
**end for**

## B    TENSOR RING EMBEDDING

Tensor Ring (TR) decomposition is a generalization to TT-decomposition where the first and the last cores are 3-dimensional tensors which corresponds to $R_0 = R_N > 1$. Formally, a tensor $\mathcal{X}$ is said to be represented in the TR format (Zhao et al., 2016) if each element of $\mathcal{X}$ can be computed as:

$$\mathcal{X}(i_1, i_2, \ldots, i_d) = \sum_{r_0=1}^{R_0} \sum_{r_1=1}^{R_1} \cdots \sum_{r_{N-1}=1}^{R_{N-1}} \mathcal{G}^{(1)}(r_0, i_1, r_1)\mathcal{G}^{(2)}(r_1, i_2, r_2) \ldots \mathcal{G}^{(N)}(r_{N-1}, i_N, r_0).$$

Similar to TT, we can define TR-matrix (see Figure 3) and corresponding TR-embedding layer.

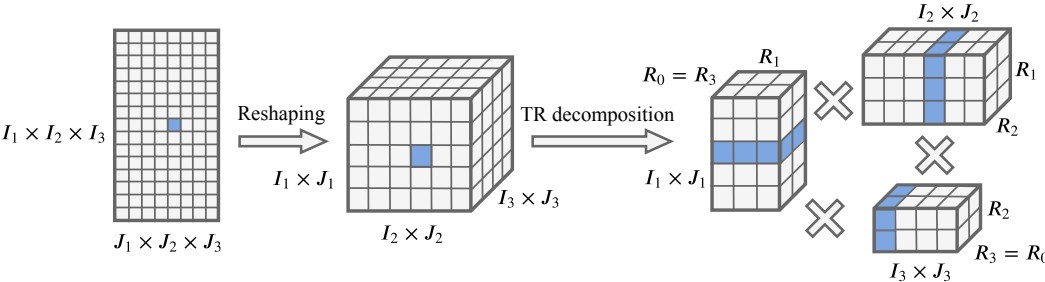

Figure 3: Construction of the TR–matrix from the standard embedding matrix. Blue color depicts how the single element in the initial matrix is transformed into the product of the highlighted matrices.

Table 5 shows the performance of different NMT models with both embedding and softmax layers replaced by either TT or TR factorizations. To achieve the same compression factor as the corresponding TT models, TR models should have smaller ranks which negatively affects their performance. Furthermore, TR is more computationally heavy.

## C    COMPLETE LIST OF HYPERPARAMETERS

Table 5: NMT, Transformer-big on WMT'14 English-to-German dataset. Both case-sensitive tokenized BLEU and de-tokenized SacreBLEU (Post, 2018) on newstest2014 are reported.

| Model | Embedding shape | TT rank | Token BLEU | Sacre BLEU | Emb compr. | Total params | Iter time |
|---|---|---|---|---|---|---|---|
| Big | $32768 \times 1024$ | — | **29.58** | **28.84** | 1 | 210M | 1.14 |
| Big+TT1 | $(32, 32, 32) \times (8, 8, 16)$ | 64 | 29.17 | 28.53 | 15.3 | 179M | 1.23 |
| Big+TT2 | $(32, 32, 32) \times (8, 8, 16)$ | 48 | 28.53 | 27.97 | 26.8 | 178M | 1.22 |
| Big+TT3 | $(32, 32, 32) \times (8, 8, 16)$ | 32 | 28.26 | 27.70 | 58.5 | 177M | 1.22 |
| Big+TR1 | $(32, 32, 32) \times (8, 8, 16)$ | 32 | 28.64 | 28.07 | 16 | 179M | 1.28 |
| Big+TR2 | $(32, 32, 32) \times (8, 8, 16)$ | 16 | 28.10 | 27.50 | 64 | 177M | 1.23 |

Table 6: Full list of hyperparameters used to train Transformer-big for NMT.

| Parameter | Value |
|---|---|
| *Data cleaning* | |
| max training sequence length in tokens | 128 |
| min training sequence length in tokens | 3 |
| max source / target difference in tokens | 25 |
| max source / target ratio | 2.5 |
| *Model* | |
| vocabulary size, $|V|$ | 32768 |
| hidden size, $d_{\text{model}}$ | 1024 |
| intermediate FF layer size, $d_{\text{ff}}$ | 4096 |
| number of attention heads, $h$ | 16 |
| number of layers in encoder / decoder | 6 |
| *Optimization* | |
| optimizer | NovoGrad |
| learning rate | 0.04 |
| betas, $(\beta_1, \beta_2)$ | $(0.95, 0.25)$ |
| learning rate decay policy | cosine |
| weight decay | 0.0001 |
| batch size in tokens | 393216 |
| number of training steps | 80000 |
| number of warmup steps | 4000 |
| *Regularization* | |
| global dropout, $P_{\text{drop}}$ | 0.2 |
| label smoothing | 0.1 |
| *Inference* | |
| beam search beam size | 4 |
| length penalty | 0 |
| max source / target difference in tokens | 50 |

Table 7: Full list of hyperparameters used to train Transformer-XL for LM.

| Parameter | Value |
|---|---|
| *Model* | |
| vocabulary size, $\|V\|$ | 267735 |
| hidden size, $d_{\text{model}}$ | 512 |
| intermediate FF layer size, $d_{\text{ff}}$ | 2048 |
| number of attention heads, $h$ | 8 |
| number of layers | 16 |
| softmax layer | adaptive |
| *Optimization* | |
| optimizer | NovoGrad |
| learning rate | 0.025 |
| betas, $(\beta_1, \beta_2)$ | $(0.95, 0.25)$ |
| learning rate decay policy | cosine |
| weight decay | 0.0001 |
| batch size in sequences | 1024 |
| target sequence length | 128 |
| memory sequence length | 128 |
| number of training steps | 150000 |
| number of warmup steps | 1500 |
| *Regularization* | |
| global dropout, $P_{\text{drop}}$ | 0.15 |
| *Inference* | |
| batch size | 4 |
| target sequence length | 128 |
| memory sequence length | 640 |
| max positional encodings length | 400 |

