# OpenReview forum: "Tensorized Embedding Layers for Efficient Model Compression"
_ICLR.cc/2020/Conference — Reject_

### Official Review · AnonReviewer2 · 2019-10-07
**Official Blind Review #2**

**Rating:** 8

**Review:**

This paper proposes a low-rank tensor decomposition model (Tensor Train-TT [Oseledets et al, 2011]) to parameterize the embedding matrix in Natural Language Processing (NLP). It shows that TT allows for a compression of the network and sometimes even a slight increase of test accuracy. The paper is well written and easy to follow.
I found the idea as a natural consequence of many recent papers proposing tensor decomposition to parameterize deep learning networks. However, I think this is the first time that the concept has been applied to learning an embedding matrix, which is an important problem in the field.
The authors reported several experimental results on different tasks and datasets for NLP such as: Sentiment Analysis, Neural Machine Translation and Language Modeling; and an application to the click through rate prediction problem.
I think the paper is of limited novelty but includes interesting experimental results that helps to better understand the potential and limitations of tensor decompositions in deep learning architectures.
Below, I summarize the issues I found, and I would like the authors to address them in their responses:
Major issues:
-	In Page 4 (and Appendix B), the authors show a comparison with Tensor Ring (TR) and conclude that TT marginally outperforms TR in terms of BLEU measure for a fixed number of parameters in both models. I found this comparison incomplete, weak and misleading because of the following reasons:
o	TR is a more general model including TT as a particular case when the first and last ranks are equal to one (Zhao et al, 2016). In fact, in this experiment, the authors chose all intermediate ranks set at the same value R with the first/last ranks set to 1 and R for TT and TR, respectively, which is not a fair comparison. Shown results suggest that first/last ranks contain less information than intermediate ranks, but it would be necessary to explore other combinations of rank values without keeping them constant to explore the generalization power of the TR model including TT as a particular case.
o	The authors compares TT and TR only in case of the NMT, Transformer-big on WMT‘14 English-to-German dataset, where the results are not good for TT and TR. It is noted that baseline model (Big) attains a Sacre BLEU = 28.84, TT1 = 28.53 and TR1 = 28.07 and the compression rate is only 210/179=1.17 for TT1 and TR1 and the Iteration time is larger than in the baseline model. In this case, there is no a clear advantage of using TT or TR.
In my opinion, to improve the paper, I think the authors could:
o	To avoid the sentence “In our experiments, however, the resulting TR embeddings performed slightly worse than TT–embeddings with the same number of parameters” unless more conclusive and exhaustive experiments are performed comparing TT and TR.
o	To add some comparison results between TT and TR for the rest of datasets such as Sentiment Analysis, Language Modeling and the click through rate prediction problem.
o	Highlight that TT is a particular case of TR so considering the first and last ranks equal to one reduce the number of parameters but can affect the generalization power of the model.
-	The approach of the paper is mostly intuitive. A theoretical result about why the low rank TT is able to catch the useful information of an optimal or suboptimal embedding matrix is missing.
Minor issues:
-	In last paragraph of section 3.2: The number of parameters is computed on the 3D tensor cores only. I think the size of the first/last 2D cores should be added. Please revise the equation.
-	The pseudocode for the mapping one index to multiple indices is trivial and could be avoided. If it is kept, I think the reverse operation should be also included, i.e. how to map multiple indices i1, …., iN to one index i.
-	The discussion and Figure 2 about the Gaussianity of the values in the higher order tensor based on Gaussian core tensors is not relevant. Maybe, the authors should better motivate why it is important to highlight that the distribution tends to a Gaussian density for increasing ranks.
-	In page 5, it is mentioned that “factors should be as close to each other as possible” but there is no a justification for it. Could you give some theoretical insight on why it is important to obtain uniform distribution of matrix size?
-	Section 4.1, reference to the Stanford sentiment treebank (SST) is missing.

On Nov 16th: I am satisfied with the responses provided by the Authors who made few changes to solve some identified minor issues. Thanks for taking the review report into account. I have raised the rating.


**Experience Assessment:**

I have published in this field for several years.

**Review Assessment: Checking Correctness Of Derivations And Theory:**

I carefully checked the derivations and theory.

**Review Assessment: Checking Correctness Of Experiments:**

I carefully checked the experiments.

**Review Assessment: Thoroughness In Paper Reading:**

I read the paper thoroughly.

---

> ### Author Response · Authors · 2019-11-11
> **On the comparison with Tensor Ring decomposition**
>
> Thank you for your time and expertise put into the review! Please, see the answers to your questions below.
>
> Major issues:
> 1. While TR is more general form of TT with powerful generalization abilities, it might require more intricate optimization to realize its full potential (Section 2.5 in [2]). We would like to stress that in the paper we do not make a conclusion of TT-embedding superiority over TR. Our experiments with TR were aimed to show one promising future direction for the work on tensorized embeddings. While our results suggest that TT-embedding shows better compression-performance trade-off than its TR counterpart on Transformer NMT, much more experimentation is needed to properly compare these two approaches. Such analysis is computationally heavy and goes beyond the scope of this paper.
> 2. The question of token embeddings optimality is difficult to answer because the very problem of determining optimality as the ability to catch useful information is ill-posed. When trained jointly in end-to-end manner, we can not separate the embedding and softmax layers from the model backbone (such as LSTM or a stack of Transformer blocks). There is some empirical evidence that reducing the number of trainable parameters in the core part will force the model to store more information in embeddings and vice versa. In particular, even models with embeddings initialized randomly and not trained at all might produce good results in some tasks [1].
>
> Minor issues:
> 1. Thank you for pointing it out, there is indeed an inaccuracy in indexing. Instead of R_k and R_{k+1} there should be R_{k-1} and R_k. Note, that the revised equation includes both 2D and 3D cores as R_0=R_N=1 by definition.
> 2. We added the pseudocode for inverse mapping into the Appendix A for completeness.
> 3. The rationale behind the discussion of Gaussianity of TT-matrix matrix elements is the following. Gaussian distribution is de facto the standard way to initialize token embeddings. It has been extensively studied and experimented with, and we know that it allows to converge to a good model if the variance is properly chosen [1]. We know very little about the right way to initialize tensorized layers, however, Figure 2 (empirically) suggests that the initialization of TT-cores with equation 2 is at least as good as the baseline if TT-rank is sufficiently high. It also shows that with the decrease of TT-rank, the initialization moves away from Gaussian which might also contribute to performance degradation (in addition to the extreme compression). We believe that this observation raises an important question and calls for a more thorough analysis of tensorized layers initialization.
> 4. Our decision for choosing as uniform shape as possible was motivated by the insights from prior work [3]. First, their big-O estimate of forward and backward passes complexity (Table1) shows that TT-layers of “imbalanced” shape are more expensive to train. Secondly, their experimental results suggest that uniform shapes perform well in contrast to TT-layers with too small number of values for particular dimensions. We hypothesize that other non-trivial shape factorizations might work better in terms of compression ratio or metric of interest. Searching for theoretical justifications of better factorizations or recipes how to choose them for each particular task is important direction for future work.
> 5. Thank you for pointing it out, we added missing reference to the updated version of the paper.
>
> [1] T. Kocmi, O. Bojar. An Exploration of Word Embedding Initialization in Deep-Learning Tasks. In ICON, 2017.
> [2] L. Grasedyck, D. Kressner, C. Tobler. A literature survey of low-rank tensor approximation techniques. CGAMM-Mitteilungen, vol. 36, pp. 53–78, 2013.
> [3] A. Novikov, D. Podoprikhin, A. Osokin, D. Vetrov. Tensorizing Neural Networks. In NIPS 2015.

---

### Official Review · AnonReviewer1 · 2019-10-23
**Official Blind Review #1**

**Rating:** 3

**Review:**

This paper introduces a novel way of parametrizing embedding layers based on the Tensor Train (TT)
decomposition, which allows compressing the model significantly at the cost of a negligible drop or even a slight gain in performance. And this paper focuses on the input embedding layers.

For the experiments, the paper just compared methods using TT layer and normal embedding layer. There are many other methods that has been proposed to compress the embedding layers, it will be good to compare with one or two other methods, such as WEST or compression based on projection layers.



**Experience Assessment:**

I have published one or two papers in this area.

**Review Assessment: Checking Correctness Of Derivations And Theory:**

I assessed the sensibility of the derivations and theory.

**Review Assessment: Checking Correctness Of Experiments:**

I assessed the sensibility of the experiments.

**Review Assessment: Thoroughness In Paper Reading:**

I read the paper thoroughly.

---

> ### Author Response · Authors · 2019-11-11
> **Comparison with other methods on embedding compression**
>
> Thank you for the review and your suggestions on additional baselines to compare with. In fact, we have some recent experimental results we plan to add to the revised version of the paper which might give an idea on the position of TT-embeddings in relation to the prior work on projection-based compression methods.
>
> After the paper submission, we encountered the ALBERT paper [1] which used simple yet efficient projection-based compression method relying on factorizing the embedding matrices as (V x D) —> (V x d) + (d x D). We replicated their setup in our NMT pipeline, however, it performed worse than TT-embedding in our experiments. For example, for ~15x embedding layer compression, TT-embedding outperforms this baseline by 0.3 SacreBLEU and this gap increases to 0.8 for ~30x compression. Further increase in the compression ratio results in even faster degradation of the baseline performance, while the performance of TT-embeddings degrades moderately. We hypothesize that this is because the baseline (as well as many other projection-based methods) decomposes the embedding matrix over the embedding dimension D only (which is usually 1-2 orders of magnitude smaller than the vocabulary size V). To achieve significant compression ratio, the embedding dimensionality has to be particularly small which might induce the Softmax bottleneck [2] and harm the model expressivity. TT-embedding, on the other hand, factorizes the embedding matrix along both vocabulary and embedding dimensions which allows to achieve higher compression ratio without significant performance degradation.
>
> Comparison to the other suggested method, Word Encoded Sequence Transducers (WEST) [3], is problematic due to the lack of its public implementation. It is also problematic to make a direct comparison with the numbers reported in the paper. The first model they experiment with, an LSTM language model (LM) trained on Penn Treebank dataset, has too high perplexity compared to what can be achieved by better optimization and regularization techniques [4]. The comparison of different methods on this benchmark is mainly a regularization game which might not fully reflect their compression properties. In our experiments on LM and NMT, we focused on SoTA architectures, significant compression of which is hard yet rewarding. The second model is also an LSTM LM which is used for automatic speech recognition (ASR) second pass rescoring and is presumably trained on the private dataset.
>
> Overall, we believe that the ease of implementation, experiments on a range of different domains, and resulting compression ratios advocate for the usability of TT-embeddings.
>
> [1] Z. Lan, M. Chen, S. Goodman, K. Gimpel, P. Sharma, R. Soricut. ALBERT: A Lite BERT for Self-supervised Learning of Language Representations. On arxiv, 2019.
> [2] Z. Yang, Z. Dai, R. Salakhutdinov, W. Cohen. Breaking the Softmax Bottleneck: A High-Rank RNN Language Model. In ICLR, 2018.
> [3] E. Variani, A. T. Suresh, M. Weintraub. WEST: Word Encoded Sequence Transducers. In ICASSP-2019.
> [4] S. Merity, N. S. Keskar, R. Socher. Regularizing and Optimizing LSTM Language Models. In ICLR, 2018.

---

### Official Review · AnonReviewer3 · 2019-10-26
**Official Blind Review #3**

**Rating:** 6

**Review:**

This paper proposes to use TensorTrain representation to transform discrete tokens/symbols to its vector representation.
Since neural networks can only work with numerical numbers, in many NLP tasks, where the raw inputs are in the discrete token/symbol format, the popular technique is to use "embedding" matrices to find a vector representation of those inputs.

As the authors point out, the embedding matrices usually require huge number of parameters, since it assigns one vector for each input token for one embedding vector, but to attain a competitive performance in the real world applications, we need to use large number of embedding vectors, which results in a large number of parameters in the neural networks.

The paper assumes that those embedding matrices can be compressed by assuming that the low-rank property of embedding matrices. I think this is a valid assumption in many cases, and the paper shows the performance degradation according to this assumption is relatively small compared to the gain, a dramatically reduced size of parameters in the embedding stage, is substantial.

I think the paper is well written and proposes a new direction to find a memory efficient representation of symbols. I am not sure the current initialization techniques, nor the training method in the paper are the right way to train a tensor train "embedding" but I expect that the authors would perform the follow up work on those topics.

**Experience Assessment:**

I have published one or two papers in this area.

**Review Assessment: Checking Correctness Of Derivations And Theory:**

I assessed the sensibility of the derivations and theory.

**Review Assessment: Checking Correctness Of Experiments:**

I assessed the sensibility of the experiments.

**Review Assessment: Thoroughness In Paper Reading:**

I read the paper at least twice and used my best judgement in assessing the paper.

---

> ### Author Response · Authors · 2019-11-11
> **TT-embedding initialization and optimization**
>
> Thank you for reviewing the paper and providing a positive feedback to our work!
>
> Our current initialization scheme was inspired by the common way to initialize token embeddings from the literature [1], which does not take into account the factorizative nature of our layers. For the experiments, we simply implemented TT-embedding and TT-softmax layers to be easily optimized along with other parameters in autodiff framework (PyTorch in our case). However, we are also aware of more advanced optimization algorithms particularly suitable for tensor decompositions (see [2] for a brief overview). There is indeed a room for improvements to our current practices of initializing and optimizing TT-embeddings which require further investigation.
>
> [1] T. Kocmi, O. Bojar. An Exploration of Word Embedding Initialization in Deep-Learning Tasks. In ICON, 2017.
> [2] A. Novikov, P. Izmailov, V. Khrulkov, M. Figurnov, I. Oseledets. Tensor Train decomposition on TensorFlow (T3F). On arxiv, 2018.

---

### Public Comment · ~Henry_Blackmore1 · 2019-10-19
**Some questions about variance and initialization**

1. What do you mean 'TT-cores are in fact 4-th order tensors' in Section 3.3? Which one is 4-th order tensor?
2. The variance of TT is said to be $R^2$ in Initialization, well, I think it's $R^N$. Can you explain this in detail or give the derivation of the Variance?
3. According to the initialization of equation (2), the variance of matrix elements follow Gaussian whatever R is, but Figure 2 does not show this. Hope you can explain this in detail
4. Can the authors also provide the baseline Tensor Ring codes for comparison?

---

> ### Author Response · Authors · 2019-10-21
> **On the TT-matrix variance and initialization, codebase update**
>
> Thanks for the interest in our paper! Please, see the answers to your questions below.
>
>  1. If a tensor of shape I1 x I2 x I3 is decomposed into a TT-format, the TT-core which corresponds to the second index is a 3-dimensional tensor of shape R1 x I2 x R2. If we deal with a TT-matrix of shape (I1 x I2 x I3) x (J1 x J2 x J3), we first reshape it into a 3-dimensional tensor of shape (I1 x J1) x (I2 x J2) x (I3 x J3) and then decompose it into a TT-format. In this case the second core is a tensor of shape R1 x (I2 x J2) x R3 which is in fact a 4-th order tensor, as now the index which corresponds to the initial tensor is 2-dimensional, not 1-dimensional.
>  2. If we understand correctly, there is a misunderstanding of what R is. In our formula, R^2 is just a designation for the variance of TT-matrix matrix element. In this case, R is a standard deviation which equals to the product of all ranks R_1, …, R_N.
>  3. Please note that only matrix elements of TT-cores follow Gaussian but not the matrix elements of the whole TT-matrix. However, matrix elements of TT-matrix approach Gaussian if we increase the TT-rank. Figure 2 shows that for small TT-ranks (<4), the distribution is rather peaky and does not resembles Gaussian, but it approaches Gaussian for bigger TT-ranks (>12).
>  4. We updated the anonymous codebase to include the latest changes to the code as well as the implementation of Tensor Ring layers.

---

### Decision · Program_Chairs · 2019-12-19

**Decision:**

Reject

**Comment:**

This paper has been reviewed by three reviewers and received scores: 6/3/8. While two reviewers were reasonably positive, they also did not provide a very compelling reviews (e.g. one rev. just reiterated the rationale behind tensor model compression and the other admitted the paper is of limited novelty). Perhaps the shortest review (and perhaps the most telling) prompts authors to the fact that the model compression with tensor decompositions is quite common in the literature these days. One example could be T-Net: Parametrizing Fully Convolutional Nets with a Single High-Order Tensor by Kossaifi et al. Very likely the authors will find many more recent developments on model compression with/without tensor decomp. For a good paper in this topic, authors should carefully consider various tensor factorizations (Tucker, TT, tensor rings, t-product and many more) and consider theoretical contributions and guarantees. Taking into account all pros and cons, this submissions falls marginally short of the ICLR 2020 threshold but the authors are encouraged to work on further developments.